# Genomic dissection of endemic carbapenem resistance reveals metallo-beta-lactamase dissemination through clonal, plasmid and integron transfer

Nenad Macesic[1,2], Jane Hawkey [1], Ben Vezina[1], Jessica A. Wisniewski [1], Hugh Cottingham[1], Luke V. Blakeway[1], Taylor Harshegyi [1], Katherine Pragastis[1], Gnei Zweena Badoordeen[1], Amanda Dennison[3], Denis W. Spelman[1,3], Adam W. J. Jenney [1,3] & Anton Y. Peleg [1,2,4] ✉

Infections caused by metallo-beta-lactamase-producing organisms (MBLs) are a global health threat. Our understanding of transmission dynamics and how MBLs establish endemicity remains limited. We analysed two decades of $bla_{IMP-4}$ evolution in a hospital using sequence data from 270 clinical and environmental isolates (including 169 completed genomes) and identified the $bla_{IMP-4}$ gene across 7 Gram-negative genera, 68 bacterial strains and 7 distinct plasmid types. We showed how an initial multi-species outbreak of conserved IncC plasmids (95 genomes across 37 strains) allowed endemicity to be established through the ability of $bla_{IMP-4}$ to disseminate in successful strain-genetic setting pairs we termed propagators, in particular *Serratia marcescens* and *Enterobacter hormaechei*. From this reservoir, $bla_{IMP-4}$ persisted through diversification of genetic settings that resulted from transfer of $bla_{IMP-4}$ plasmids between bacterial hosts and of the integron carrying $bla_{IMP-4}$ between plasmids. Our findings provide a framework for understanding endemicity and spread of MBLs and may have broader applicability to other carbapenemase-producing organisms.

Carbapenemase-producing organisms (CPOs) are now endemic in many regions. While there has been a significant focus on $bla_{KPC}$ due to its spread through North America and Europe, metallo-beta-lactamases (MBLs) (e.g. $bla_{NDM}$, $bla_{IMP}$ and $bla_{VIM}$) are endemic through much of Asia and Oceania including Australia, where $bla_{IMP}$ carbapenemases have dominated[1–9]. Treatment options for infections caused by CPOs, particularly MBL-harbouring organisms, remain severely limited, highlighting the need to stop further spread of these extensively drug-resistant organisms. The mechanisms for carbapenemase spread differ according to carbapenemase type. Some carbapenemases spread through close associations with successful strains or lineages (e.g. $bla_{KPC-2/3}$ and *Klebsiella pneumoniae* clonal complex 258), while for others, spread is mediated through association with specific plasmids (e.g. $bla_{OXA-48}$ and broad-host range IncL plasmids)[10,11]. Notably, MBLs spread through both lineage-related clonal expansion and diverse plasmid types[12,13]. While surveillance studies have captured some of these data, there have been few efforts to assess how these mechanisms of spread evolve over time. Understanding the transmission dynamics of carbapenem resistance genes will be crucial to inform future infection prevention efforts.

[1]Department of Infectious Diseases, The Alfred Hospital and Central Clinical School, Monash University, Melbourne, Australia. [2]Centre to Impact AMR, Monash University, Clayton, Australia. [3]Microbiology Unit, Alfred Hospital, Melbourne, Australia. [4]Infection Program, Monash Biomedicine Discovery Institute, Monash University, Clayton, Australia. ✉e-mail: anton.peleg@monash.edu

Previous work from our group and others have identified that the spread of MBLs, and particularly $bla_{\text{IMP-4}}$, is often driven by dissemination by transposons of a class 1 integron that has been able to insert into several genetic settings (henceforth defined as either chromosomal integration or different plasmid types carrying $bla_{\text{IMP-4}}$)[2,4,6,9,14,15]. Furthermore, the gene cassettes (such as that carrying $bla_{\text{IMP-4}}$) may also be a source of dissemination by being able to enter different class 1 integrons[16]. The ability to study horizontal gene transfer has been significantly advanced by long-read sequencing, which enables high-quality de novo assembly of bacterial genomes, including highly repetitive regions such as plasmids. Utilising long-read sequencing to generate completed, closed bacterial genomes provides a unique opportunity to study the complex, multi-level (bacterial strain, plasmid, gene) transmission dynamics that are likely occurring during MBL spread. When combined with short-read sequencing data, an unprecedented level of detail of the genetic context and likely mechanisms of an outbreak or endemicity is possible.

MBL-producing Gram-negative bacteria, dominated by $bla_{\text{IMP-4}}$, have been isolated in our institution (The Alfred Hospital) since 2002[5–7,15]. After an initial outbreak period in 2004–2005, we experienced hyperendemicity, with a repeated outbreak period from 2017–2020. We aimed to assess the genetic settings of $bla_{\text{IMP-4}}$, its evolution over time, and the transmission pathways that resulted in repeated outbreaks and endemicity.

In this work we used long- and short-read whole genome sequencing to characterise the genetic settings of $bla_{\text{IMP-4}}$ in bacterial chromosomes and plasmids from 277 clinical and environmental isolates from 2002–2020. This allowed us to track the spread of $bla_{\text{IMP-4}}$ in 7 plasmid types and multiple chromosomal settings. In the 18-year period, we noted incredible plasticity of $bla_{\text{IMP-4}}$ persistence, with vertical spread through the transmission of dominant strains and horizontal spread of both plasmids and the class 1 integron carrying $bla_{\text{IMP-4}}$. We also identified a persistent reservoir of $bla_{\text{IMP-4}}$ in IncC plasmids in both clinical and environmental isolates. Our findings highlight the need for integration of long-read sequencing into CPO surveillance, as well as for multi-modal infection prevention approaches that address the diverse forms of CPO spread.

## Results

### $bla_{\text{IMP-4}}$ found in diverse clinical and environmental isolates spanning two decades

We sequenced 277 $bla_{\text{IMP-4}}$-harbouring isolates from an institutional collection of carbapenem-resistant isolates systematically collected from 2002–2020, including 264 clinical isolates from 196 patients and 13 environmental isolates (Supplementary Dataset 1). This included short-read (Illumina) data on all isolates and long-read (Oxford Nanopore) data on 172 isolates that best represented the strains across the study time periods. Seven isolates failed quality control and were excluded. In total, we analysed 270 isolates that were made up of 68 bacterial strains (defined as unique species/multi-locus sequence type [MLST] combinations) from 7 Gram-negative genera, highlighting the diversity of bacterial hosts for $bla_{\text{IMP-4}}$ (Fig. 1a). The five most frequent strains accounted for 190/270 (70%) genomes and included *Serratia marcescens* (52/270 isolates, 19%), *Enterobacter hormaechei* ST190 (44/270 genomes, 17%), *E. hormaechei* ST93 (36/270 genomes, 13%), *Pseudomonas aeruginosa* ST111 (35/270 genomes, 13%) and *E. hormaechei* ST114 (23/270 genomes, 9%) (Fig. 1a). In addition to $bla_{\text{IMP-4}}$, 8/270 (3%) genomes carried other carbapenemase genes (4 $bla_{\text{OXA-58}}$, 2 $bla_{\text{NDM-7}}$, 1 $bla_{\text{NDM-1}}$, 1 $bla_{\text{KPC-2}}$, 1 $bla_{\text{OXA-500}}$) and 121/270 (44%) carried *mcr-9.1*, a novel determinant of colistin resistance[14] (Supplementary Dataset 1).

### $bla_{\text{IMP-4}}$ detected in multiple plasmid and chromosomal genetic settings across three distinct time periods

We first determined the genetic setting of $bla_{\text{IMP-4}}$ using 169 completed, circularised genomes (three non-circularised genomes were excluded). $bla_{\text{IMP-4}}$-carrying plasmids were clustered using MOB-typer[17], which uses a whole-sequence-based typing system to provide cluster codes for reconstruction and tracking of plasmids. Representative plasmids from each cluster were then used as references for mapping of the 99 genomes with short-read data only and the three non-circularised genomes (Supplementary Dataset 1). Overall, 230 and 40 isolates carried $bla_{\text{IMP-4}}$ on a plasmid or on the chromosome, respectively, with seven distinct plasmid types identified and chromosomal integration in multiple strains (Fig. 1b and Supp. Table 1). No genomes showed concurrent chromosomal integration and plasmid carriage of $bla_{\text{IMP-4}}$. For the majority of isolates (151/169, 89%), $bla_{\text{IMP-4}}$

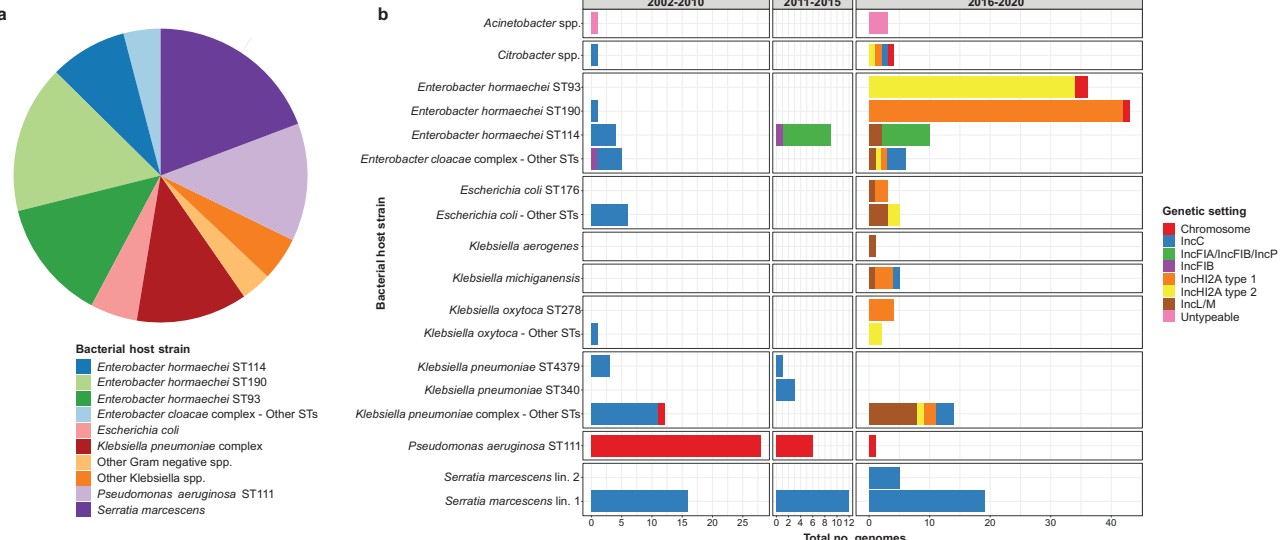

**Fig. 1 | Summary of bacterial host species, multi-locus sequence types and genetic settings of $bla_{\text{IMP-4}}$ in sequenced isolates. a** Pie chart showing key bacterial host strains of $bla_{\text{IMP-4}}$. $bla_{\text{IMP-4}}$ was noted in 7 bacterial genera and 68 bacterial host strains. **b** Genetic settings and bacterial host species of $bla_{\text{IMP-4}}$ over course of study, as defined by three distinct periods. Abbreviations: No number, Lin lineage, ST sequence type.

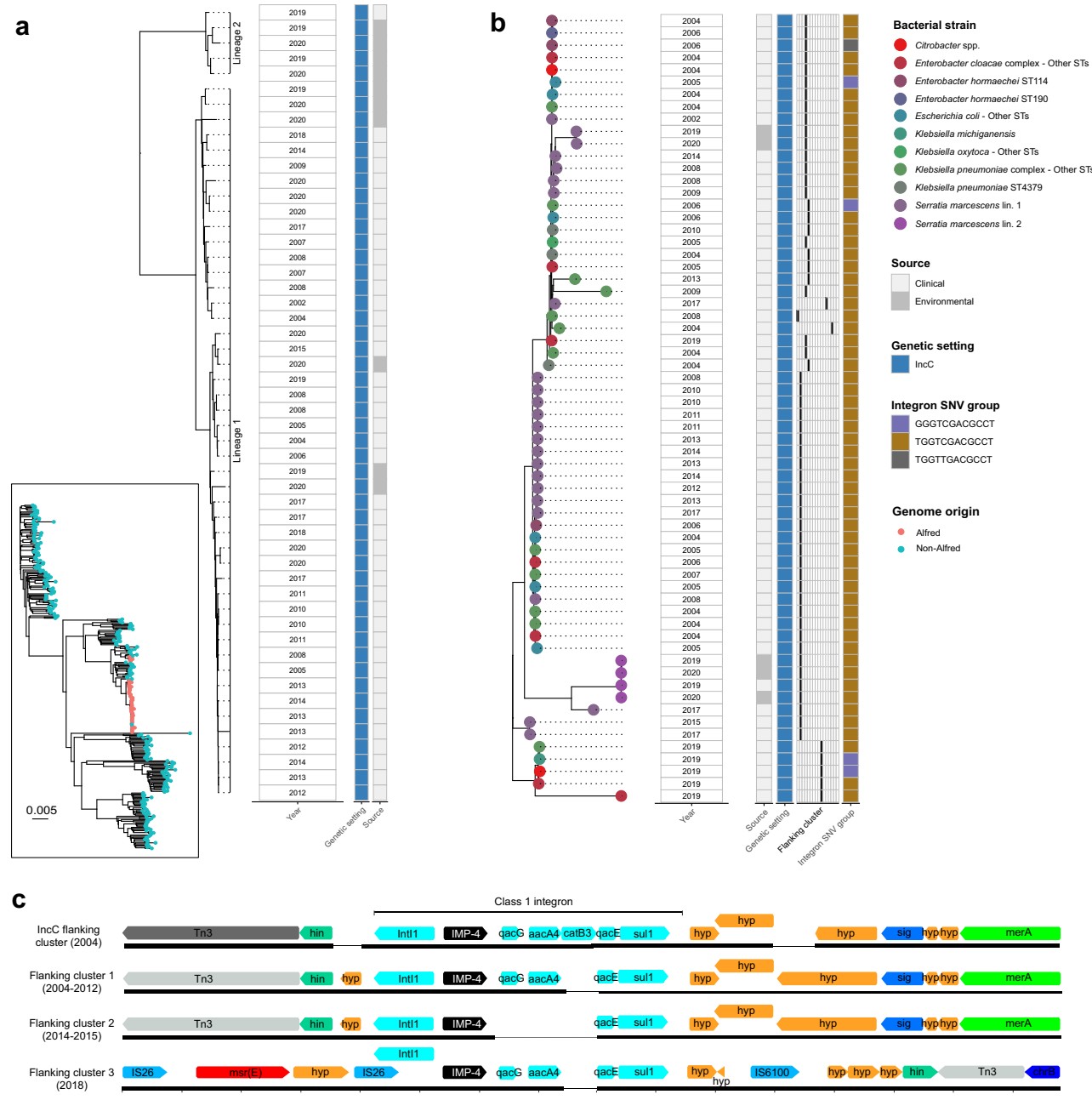

**Fig. 2 | Phylogenetic analysis of $bla_{IMP-4}$-carrying *S. marcescens* and IncC plasmids and analysis of $bla_{IMP-4}$ flanking regions in *Pseudomonas aeruginosa* ST111. a** Phylogenetic analysis of $bla_{IMP-4}$-carrying *S. marcescens* genomes from Alfred Hospital. Inset panel locates *S. marcescens* genomes from the Alfred Hospital in a global *S. marcescens* phylogeny. Outer panel shows Alfred Hospital genomes only, indicating that they formed two distinct lineages and were detected both in clinical and environmental samples. **b** Phylogenetic analysis of $bla_{IMP-4}$ IncC plasmids using Mashtree. IncC plasmids entered 37 bacterial host strains but remained stable with four flanking region clusters and one integron SNV profile accounting

for 93% and 90% plasmids, respectively. **c** Analysis of $bla_{IMP-4}$ flanking regions in *P. aeruginosa* ST111. Three flanking clusters were detected in *P. aeruginosa* ST111. Differences are shown to the class 1 integron present in contemporaneous $bla_{IMP-4}$ IncC plasmids (top), with absence of *catB3* cassette in flanking cluster 1 (in genomes from 2004–2012), absence of *qacG, aac4* and *catB3* cassettes in flanking cluster 2 (2014–2015) and distinct flanking regions and absence of *catB3* cassette in flanking cluster 3 (2018). Abbreviations: bp base pairs, Lin. lineage, SNV single nucleotide variant, ST sequence type.

was situated in a class 1 integron most commonly comprising the $bla_{IMP-4}$-*qacG-aacA4-catB3-qacE-sul1* cassette array. The bacterial host strain-plasmid relationships evolved over the course of the study, with three distinct time periods (Fig. 1b).

**Outbreak initiation and establishment of $bla_{IMP-4}$ endemicity**
$bla_{IMP-4}$ was first noted in a clinical *S. marcescens* isolate in 2002, with $bla_{IMP-4}$ being carried on an IncC plasmid (Supp. Fig. 1). It took ~2 years

before further $bla_{IMP-4}$-carrying Gram-negative bacteria were identified, and these were dominated by an IncC genetic setting or chromosomal $bla_{IMP-4}$ in *P. aeruginosa* (Fig. 1b). Notably, the first *S. marcescens* lineage (here called lineage 1) with IncC-carrying $bla_{IMP-4}$ became a successful lineage across the entire study period (Fig. 1b). *S. marcescens* lineage 1 included 47 genomes from both clinical and environmental (intensive care unit [ICU] sinks) isolates. Based on their genetic relatedness (median pairwise single nucleotide variant [SNV]

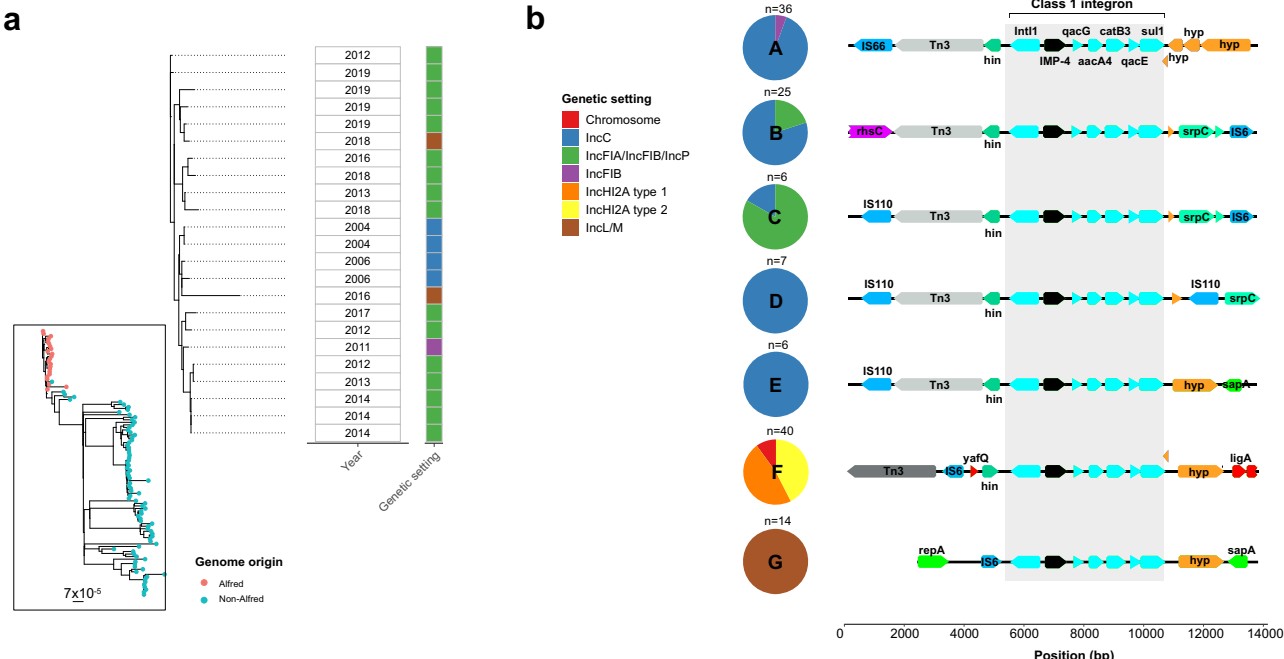

**Fig. 3 | Phylogenetic analysis of $bla_{IMP-4}$-carrying *E. hormaechei* ST114 and clustering analysis of $bla_{IMP-4}$-flanking regions. a** Phylogenetic analysis of $bla_{IMP-4}$-carrying *E. hormaechei* ST114 genomes from Alfred Hospital. Inset panel places *E. hormaechei* ST114 genomes from the Alfred Hospital in a global *E. hormaechei* ST114 phylogeny. Outer panel shows Alfred Hospital genomes only. Although clonally related, they were more diverse than other key $bla_{IMP-4}$ bacterial host strains and were able to act as versatile acceptors of $bla_{IMP-4}$ in different genetic settings,

including four different plasmid types throughout the study. **b** Clustering analysis of flanking regions 5000 bp upstream and downstream of $bla_{IMP-4}$ using Flanker. The six key flanking region clusters with >5 genomes are shown on the left and the genetic context is shown on the right. IncFIB and IncFIA/IncFIB/IncP plasmids clustered with IncC plasmids throughout the analysis and shared homology both upstream and downstream of $bla_{IMP-4}$. Abbreviations: bp. base pairs, SNV single nucleotide variant.

distance of 8 [IQR 4–11]), vertical transmission was most likely (Fig. 2a and Supp. Table 2). The IncC plasmid was rapidly noted in diverse bacterial hosts, with expansion from *S. marcescens* into 13 other bacterial strains in 2004 alone. Ultimately, this plasmid was noted in 95 genomes across 37 strains (Fig. 2b), with *S. marcescens* genomes accounting for the majority (52/95 [54.7%]). Despite the diversity of IncC-carrying bacterial hosts over the 18 years, IncC plasmids were highly conserved, with only 11 SNVs across the plasmid backbone (median pairwise distance of 0 SNVs) (Supp. Table 3) and almost all (70/71, 98.5%) IncC plasmids belonged to the same MOB-typer primary/secondary clusters (AA860 AJ266, Supp. Table 1). We noted seven IncC mosaic plasmids with additional replicon types that were excluded from further clustering analyses (Supp. Table 4).

We used Mashtree[18] to further characterise relatedness of the 64 non-mosaic IncC plasmids, which highlighted the similarity of plasmids across bacterial hosts and different time periods (Fig. 2b). This was also reflected in the $bla_{IMP-4}$ flanking regions and integron sequences. Cluster analysis of the flanking regions up to 5000 bp upstream and downstream of $bla_{IMP-4}$ using Flanker[19] identified only four flanking regions (excluding singletons) in IncC plasmids, accounting for 67/71 (94%) genomes (Fig. 2b). The $bla_{IMP-4}$ containing integrons in the IncC plasmids were also highly similar, with a single SNV profile (TGGTCGACGCCT) accounting for 63/69 (91%) plasmids with intact integrons (Fig. 2b). Taken together, these findings suggest that $bla_{IMP-4}$ containing IncC plasmids dominated the early outbreak period and established endemicity through their ability to rapidly spread across different bacterial hosts while maintaining stability. *S. marcescens* was a persistent host and reservoir for $bla_{IMP-4}$ IncC during this time period (2002–2010) (Supp. Fig. 1).

In addition to $bla_{IMP-4}$ IncC plasmids, chromosomal integration of $bla_{IMP-4}$ into *P. aeruginosa* ST111 (a global MDR lineage)[20] was also a dominant feature of this early time period (Fig. 1b and Supp. Fig. 1).

These isolates were rapidly noted in nine patients in 2004 and continued to be isolated until 2018. The pseudomonal isolates were highly related (mean pairwise SNV distance of 1.6 SNVs vs 38.7 SNVs between Alfred and publicly available ST111 genomes [$P < 0.001$]) (Supp. Fig. 2a) but the $bla_{IMP-4}$ containing integron and the flanking regions differed depending on the time period of isolation, and were also different to the integron and flanking region sequence of the IncC plasmids (Fig. 2c). Despite being temporally associated, the chromosomal integration of $bla_{IMP-4}$ in *P. aeruginosa* ST111 with a different integron structure and flanking regions suggests that $bla_{IMP-4}$ entry into *P. aeruginosa* likely arose independently of the $bla_{IMP-4}$ IncC plasmids.

### Low endemicity of $bla_{IMP-4}$ and entry into novel plasmids

Apart from ongoing isolation of $bla_{IMP-4}$ IncC plasmids (predominately in *S. marcescens*) and chromosomal $bla_{IMP-4}$ *P. aeruginosa*, the evolution of the following time period (2011–2015) was characterised by $bla_{IMP-4}$ entering novel plasmids in *E. hormaechei* ST114 (a global MDR nosocomial *Enterobacter* clone)[21] (Fig. 1b and Supp. Fig. 1). At the start of the outbreak, there were a small number of *E. hormaechei* ST114 with $bla_{IMP-4}$-carrying IncC plasmids, but during this period, *E. hormaechei* ST114 acquired $bla_{IMP-4}$-carrying IncFIB and IncFIA/IncFIB/IncP plasmids (Fig. 1b). Phylogenomic analysis showed that the *E. hormaechei* ST114 isolates were more diverse (median pairwise SNV distance 35, IQR 27–46) but the Alfred Hospital isolates still clustered more closely than other publicly available genomes (Fig. 3a and Supp. Table 2). To determine if the IncC plasmids were the source of $bla_{IMP-4}$ in the IncFIB and IncFIA/IncFIB/IncP plasmids, we analysed the $bla_{IMP-4}$ flanking regions across the three plasmids (Fig. 3b). We noted homology of the 3850 bp upstream and 305 bp downstream regions of the integron in IncC, IncFIA/IncFIB/IncP and IncFIB plasmids (clusters A, B and C) with Tn3 transposons and DNA recombinases (*hin*) located immediately upstream.

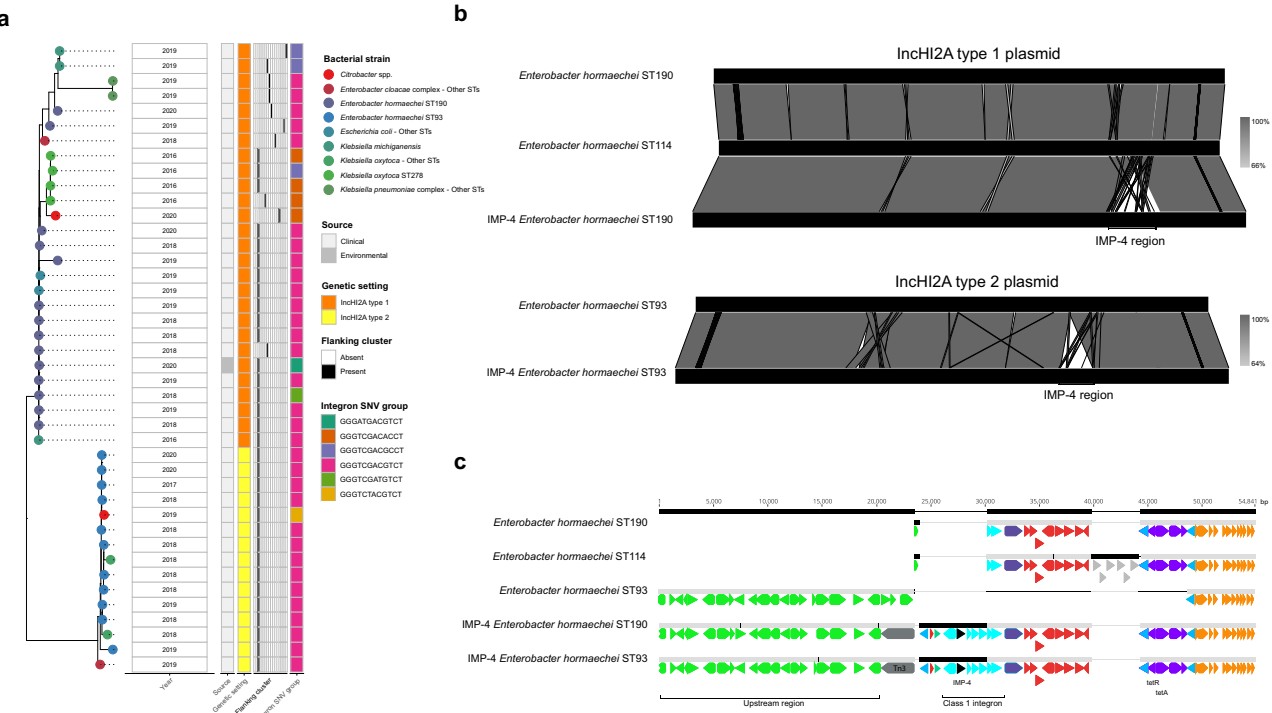

**Fig. 4 | Phylogenetic analysis of *bla*$_{IMP-4}$-carrying IncHI2A plasmids and comparison to non-*bla*$_{IMP-4}$ IncHI2A plasmids. a** Phylogenetic analysis of IncHI2A plasmids using Mashtree. Two distinct plasmid types (IncHI2A type 1 and type 2) were noted from 2016–2020 and rapidly entered 15 bacterial strains. These plasmids shared flanking regions and integron SNV profiles, which were distinct to those noted in other plasmid types. **b, c** Comparative analysis of *bla*$_{IMP-4}$-carrying IncHI2A plasmids to non-*bla*$_{IMP-4}$ IncHI2A plasmids from *E. hormaechei* ST190, ST93 and ST114 from Alfred Hospital. IncHI2A type 1 plasmids from *E. hormaechei* ST190 had homology over 100% of the non-*bla*$_{IMP-4}$ plasmid, with the addition of a 33 kbp region carrying the class 1 integron in the *bla*$_{IMP-4}$ plasmid. IncHI2A type 2 plasmids from *E. hormaechei* ST93 shared homology across 97.6% of the non-*bla*$_{IMP-4}$ plasmid, with the *bla*$_{IMP-4}$ integron contained in a mosaic region. Abbreviations: bp base pairs, SNV single nucleotide variant, ST sequence type.

(Fig. 3b). These three plasmids also shared the same integron SNV profile (TGGTCGACGCCT) (Supp. Fig. 3). Taken together, these findings suggested that as the *bla*$_{IMP-4}$-harbouring IncC plasmids became endemic during the first time period (2002–2010), the outbreak evolved whereby the IncC plasmids served as a *bla*$_{IMP-4}$ reservoir not only for inter-strain plasmid transfer, but also transfer of the *bla*$_{IMP-4}$ integron and flanking regions between IncC, IncFIA/IncFIB/IncP and IncFIB plasmids.

### Hyperendemicity and repeat outbreaks driven by clonal expansion of new bacterial strains and inter-species plasmid spread

The most recent time period (2016–2020) was characterised by complex, multi-level transmission dynamics resulting from the emergence of several new and highly successful *bla*$_{IMP-4}$ plasmids and clonal expansion of *E. hormaechei* ST190 and ST93 host strains (Fig. 1b). Ongoing circulation of *bla*$_{IMP-4}$ in genetic settings and bacterial strains from prior periods was also observed (Fig. 1b). Early in this period, *bla*$_{IMP-4}$ was identified in a new plasmid, IncL/M, first in *E. hormaechei* ST114, which was its fourth *bla*$_{IMP-4}$ carrying plasmid, and then in a wide range of other bacterial strains (*n* = 16) (Fig. 1b). All IncL/M plasmids belonged to the same MOB-typer cluster, shared the same SNV profile in the integron (GGGTCGACGCCT) and 14/17 shared the same flanking cluster (Cluster G) (Fig. 3b). The three plasmids with other flanking clusters had minor variations in the Cluster G flanking region leading to them being clustered as singletons. These flanking regions were distinct from all other *bla*$_{IMP-4}$ plasmids, however the same integron SNV profile was also noted in other plasmids (Supp. Table 5 and Supp. Fig. 3).

In 2017, *bla*$_{IMP-4}$ was detected for the first time in IncHI2A plasmids in a small outbreak of *Klebsiella oxytoca* ST278 and *Klebsiella michiganensis* ST50 (Fig. 1b). The first IncHI2A plasmid (type 1—as defined by MOB-typer cluster AA739 AJ055) then spread to *E. hormaechei* ST190 and a second IncHI2A plasmid (type 2—MOB-typer cluster AA739 AJ058) emerged in *E. hormaechei* ST93, with both bacterial strains undergoing significant clonal expansion and contributing to a repeated outbreak and hyperendemicity from 2017–2020 (Figs. 1b and 4a). Bacterial isolates carrying *bla*$_{IMP-4}$ on these two IncHI2A plasmids ultimately accounted for 98/161 (61%) of the sequenced genomes in that period (with 36/98 *E. hormaechei* ST93 and 43/98 *E. hormaechei* ST190) (Fig. 4a). The *E. hormaechei* ST93 and ST190 bacterial hosts were highly clonal with a median pairwise SNV distance of 9 (IQR 2-14) and 3 (IQR 2–4), respectively (Supp. Fig. 2b and 2c, Supp. Table 2). In addition to these two strains, the IncHI2A plasmids were found in 13 other strains (Fig. 4a). Analysis of the *bla*$_{IMP-4}$ flanking regions and integrons in the IncHI2A plasmids showed the same flanking sequence (Cluster F) across 36/47 (77%) plasmids and the same integron SNV profile (GGGTCGACGTCT) in 35/47 (74%) plasmids across both IncHI2A plasmid types (Fig. 4a). These flanking sequences and integron SNV profiles were not found in other plasmid types (Supp. Fig. 3), suggesting that they may have arisen independently of other *bla*$_{IMP-4}$ genetic settings.

To understand the rapid appearance of *bla*$_{IMP-4}$ IncHI2A plasmids, we compared them to carbapenem-susceptible, non-*bla*$_{IMP-4}$ IncHI2A plasmids in single *E. hormaechei* ST114, ST190 and ST93 genomes from our institution. The plasmids were highly similar between *bla*$_{IMP-4}$ and non-*bla*$_{IMP-4}$ bacterial strains (Fig. 4b). The IncHI2A type 1 plasmids from *bla*$_{IMP-4}$ *E. hormaechei* ST190 had the addition of a 33 kbp region

**a**

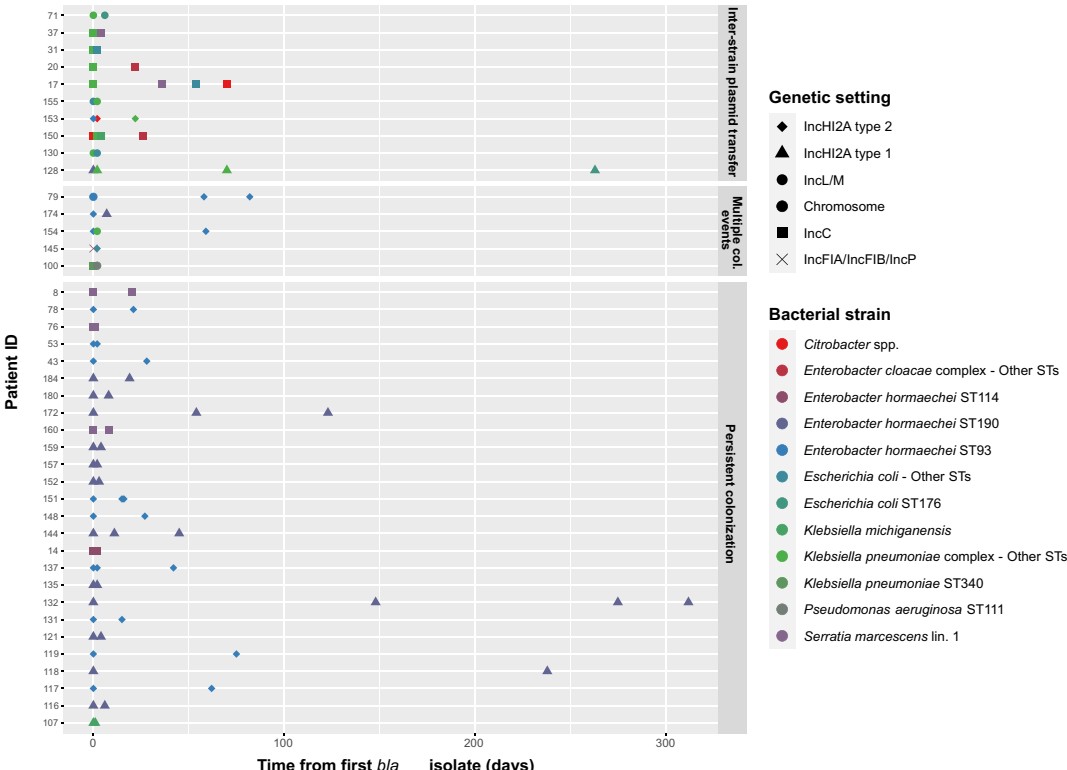

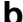

**b**

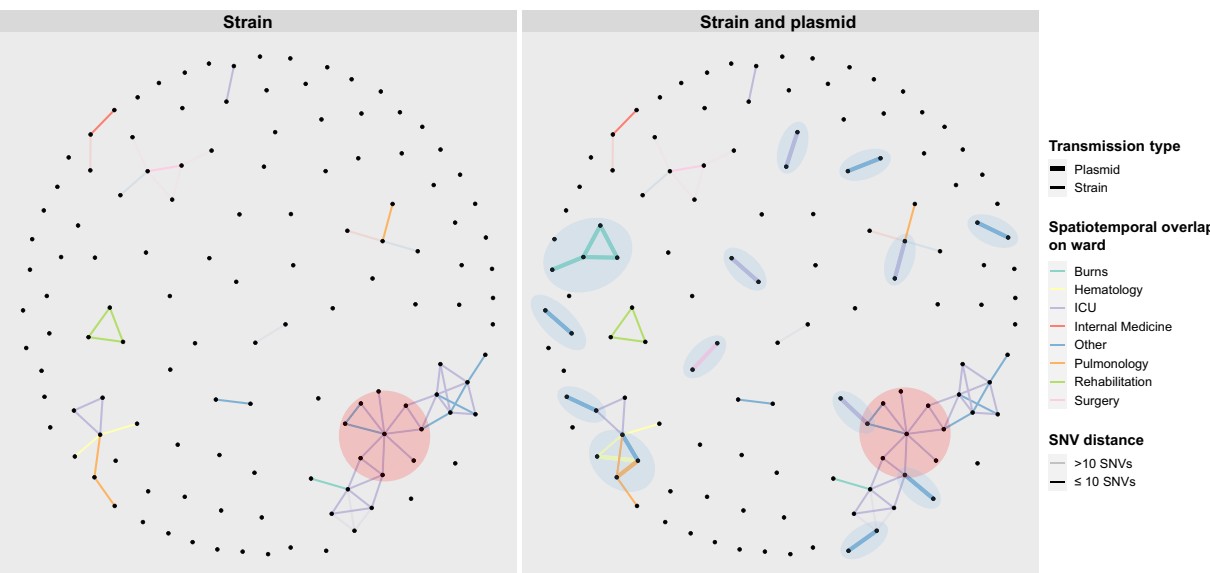

**Fig. 5 | Within-patient longitudinal analysis of** *bla*IMP-4 **spread and between-patient** *bla*IMP-4 **transmission analysis. a** Longitudinal analysis of *bla*IMP-4 in patients with multiple genomes available. Each line represents a patient with colours representing the bacterial host and shapes representing the *bla*IMP-4 genetic setting. **b** Analysis of potential transmission events between patients. Individual patients are shown as vertices. Edges were drawn if there was spatiotemporal overlap on a hospital wards (indicated by edge colour) and a genomic criterion was met. In the left panel, the genomic criterion was having the same *bla*IMP-4 bacterial strain with SNV distance from phylogenetic analysis indicated by shading. In the right panel, the genomic criterion incorporated both the strain transmission analysis as well as detecting presence of *bla*IMP-4 in the same plasmid (as defined by MOB-typer cluster) in different bacterial strains. The additional benefit of plasmid analysis is indicated by the blue shading of the potential transmission events in this panel. The red shading indicates the patient with the highest betweenness-centrality and their contacts. Abbreviations: ICU intensive care unit, Lin. lineage, SNV single nucleotide variant, ST sequence type.

carrying the class 1 integron with $bla_{IMP-4}$ (Fig. 4b). The IncHI2A type 2 plasmids from $bla_{IMP-4}$ *E. hormaechei* ST93 shared 97.6% homology with the non-$bla_{IMP-4}$ plasmid, with the $bla_{IMP-4}$ integron contained in a mosaic region (Fig. 4b, c). These data suggested integration of $bla_{IMP-4}$ into pre-existing carbapenem-susceptible *Enterobacter* carrying IncHI2A plasmids, with mobilisation of the regions upstream and downstream. These flanking regions in IncHI2A plasmids were distinct to those in other $bla_{IMP-4}$ plasmids, with a different IS110-like insertion sequence/Tn3 transposon combination (Fig. 4c).

While the arrival of new IncHI2A and IncL/M plasmids was the major contributor to the repeated outbreak and hyperendemicity during this period, we saw ongoing circulation of $bla_{IMP-4}$ in prior genetic settings. In particular, we saw IncC plasmids circulating (32/161 genomes, 19.9%), including in a novel lineage of *S. marcescens* in 2019 (lineage 2—Fig. 2a). We also noted $bla_{IMP-4}$-harbouring *P. aeruginosa* ST111 (1 genome) and *E. hormaechei* ST114 with $bla_{IMP-4}$ IncFIA/IncFIB/IncP plasmids (9 genomes). This reflected a cumulative trend where the prior $bla_{IMP-4}$ genetic settings persisted in the context of new bacterial strains and plasmids, rather than waxing and waning over time.

### An environmental reservoir of $bla_{IMP-4}$ plasmids
In response to the outbreak in the most recent time period, we conducted environmental screening of ICU sinks from 2019–2020 and cultured 34 $bla_{IMP-4}$ isolates (33 *S. marcescens* and one *E. hormaechei*), with 11 isolates selected for sequencing. Despite *E. hormaechei* predominating in $bla_{IMP-4}$ clinical isolates, 6 genomes were *S. marcescens* lineage 1, 4 *S. marcescens* lineage 2, and one *E. hormaechei* ST190. These genomes closely matched clinical isolates, with $bla_{IMP-4}$ being located on IncC plasmids and IncHI2A type 1 plasmids in *S. marcescens* and *E. hormaechei* ST190, respectively (Figs. 2b and 4a). This indicated that sinks were a possible reservoir for $bla_{IMP-4}$ and may have explained the persistence of *S. marcescens* with closely matching IncC plasmids throughout the study (Fig. 2b).

### Within-patient and between-patient $bla_{IMP-4}$ analyses show importance of diversification of genetic settings through plasmid transfer
Patients with serial $bla_{IMP-4}$-harbouring genomes available followed different trajectories of $bla_{IMP-4}$ carriage (Fig. 5a). Multiple colonisation events were noted in 5/41 patients, with $bla_{IMP-4}$ being located in distinct genetic settings (i.e. differing plasmids and/or chromosomal integration). Flanking region/integron SNV profiles also differed, making within-patient integron transfer unlikely. Possible within-patient inter-strain transfer of key $bla_{IMP-4}$ plasmids (IncC, IncL/M, IncHI2A types 1 and 2) occurred in 10/41 patients with the same $bla_{IMP-4}$ plasmid types being noted in multiple strains (Fig. 5a). In 7 patients, the evidence was particularly compelling as plasmids had identical flanking sequences and integron SNV profiles across different bacterial hosts. Persisting colonisation was noted in 26 patients, with the same strain and same $bla_{IMP-4}$ genetic setting repeatedly isolated.

We then used patient movement data (available in 127 patients from 2013 onwards) to establish putative transmission events (Fig. 5b), defined as spatiotemporal overlap between patients and genomic evidence of potential transmission. For genomic evidence, we considered both strain transmission (same bacterial host strain carrying $bla_{IMP-4}$ in the same genetic setting) and plasmid transmission (detection of the same plasmid by MOB-typer cluster in different bacterial strains). We linked 71/127 (56%) patients using these definitions and identified the ICU as a major transmission site with 36/76 (47%) potential transmission events across 7/16 (44%) transmission networks, including the two largest networks (23 and 9 patients, respectively). While strain transmission contributed significantly, use of long-read sequencing to detect potential plasmid transmission allowed us to detect an additional 5/10 (50%) wards, 7/16 (44%)

transmission networks and link 22/71 (31%) patients beyond what was identified for strain transmission alone (Fig. 5b). We measured betweenness centrality to identify key patients involved in transmission[22]. The patient with the highest betweenness centrality (164.0 vs mean 4.9) had a > 4-month ICU admission and was implicated in 7 transmission events, placing them at the centre of the large 23-patient network spanning those two strains and plasmid types (*E. hormaechei* ST190 with IncHI2A type 1 plasmid and *E. hormaechei* ST93 with IncHI2A type 2 plasmid) (Fig. 5b). A further transmission event to a single patient then occurred during an admission on the Cardiology ward 6 months later.

## Discussion
The spread of carbapenemases is the major driver of carbapenem resistance globally[23] and has been the focus of numerous cross-sectional studies[12,13,24]. To date, there have been limited efforts to study carbapenem resistance over extended time periods[25,26]. In this study we had a unique opportunity to analyse two decades of $bla_{IMP-4}$ carbapenemases in our institution and gained important insights into how $bla_{IMP-4}$ caused outbreaks and perpetuated endemicity. $bla_{IMP-4}$ spread occurred through multiple mechanisms including strain transmission, plasmid transmission and transfer of the $bla_{IMP-4}$ class 1 integron. Each of these had a different qualitative and quantitative contribution to $bla_{IMP-4}$ persisting at our institution, highlighting that endemicity is a nuanced process requiring these mechanisms to act in concert. These findings carry important implications for prevention of future carbapenemase endemicity.

From a pathogen perspective, we propose that there are two key conditions required for $bla_{IMP-4}$ endemicity. Firstly, there is a need for diversification of genetic settings for the resistance determinant, which in our study occurred through extensive inter-strain transmission of key $bla_{IMP-4}$ plasmids (IncC, IncHIA2 type 1 and type 2, IncL/M), as well as mobilisation by transposons of the class I integron and entry into new plasmids (IncFIB, IncFIA/IncFIB/IncP). This ability to diversify led to the initial establishment of endemicity with IncC plasmids, and also to the period of hyperendemicity due to the emergence of a novel context in IncHI2A and IncL/M plasmids. To study this emergence, we demonstrated that $bla_{IMP-4}$ IncHI2A plasmids were highly similar to non-$bla_{IMP-4}$ IncHI2A plasmids in *E. hormaechei* ST93, ST114 and ST190 that may have served as acceptors for the $bla_{IMP-4}$ integron. In addition, there was possible importation from outside sources: $bla_{IMP-4}$ has been found in both IncHI2A and IncL/M plasmids in Australian isolates[2,4,9,27] and $bla_{IMP-4}$-harbouring IncHI2A plasmids are emerging as a global issue, having been noted in a recent multi-hospital outbreak in the United Kingdom[28].

The second condition is propagation of $bla_{IMP-4}$ through the establishment of high-risk strain-genetic setting pairs we term propagators. While we noted $bla_{IMP-4}$ in 68 strains during the study, five strains accounted for 190/270 (70%) genomes and 140/196 (71%) patients colonised with $bla_{IMP-4}$. The first of these was *S. marcescens* lineage 1 (IncC plasmids), which defined the early period of the study and continued to persist throughout. This propagator pair was able to act as a reservoir of $bla_{IMP-4}$, likely through occupying an environmental niche as we noted during sampling of ICU sinks. The colonisation of hospital plumbing by CPOs has been well documented[9,29–32], including $bla_{IMP-4}$-harbouring *S. marcescens* in an Australian setting that was unable to be eradicated[33]. This environmental colonisation probably enabled clonal spread of $bla_{IMP-4}$-harbouring *S. marcescens* and may have driven diversification through inter-strain transfer of $bla_{IMP-4}$ IncC plasmids and inter-plasmid transfer of the $bla_{IMP-4}$ class I integron. Other propagators emerged at various junctures including *P. aeruginosa* ST111 (chromosome) and *E. hormaechei* ST114 (IncFIA/IncFIB/IncP plasmids), then *E. hormaechei* ST190 and ST93 (IncHI2A type 1 and type 2 plasmids, respectively). Clonal spread of propagators was therefore central to establishing and maintaining $bla_{IMP-4}$

endemicity, as well as leading to a repeated outbreak and hyper-endemicity in the final period of the study. These findings broadly fit the 'multiple lineages, multiple plasmids' designation proposed by David et al. when analysing carbapenemase spread in *K. pneumoniae*[12] but we demonstrated that the dynamics of carbapenemase endemicity in our setting were significantly more complex with clonal transmission of propagator strains, inter-strain plasmid transmission and inter-plasmid integron transmission all playing important roles.

In addition to pathogen factors, we were able to analyse patient factors. While genomic surveillance previously focused on lineage-level analysis, long-read sequencing technologies have improved analysis of plasmids and other mobile genetic elements[12,28,34]. In our study, these insights proved informative both for understanding within-patient and between-patient $bla_{IMP-4}$ spread. Within patients, we detected different trajectories of colonisation. Patients who undergo multiple colonisation events may be at the core of multiple transmission networks, as demonstrated by the patient colonised with two IncHI2A plasmids and highest betweenness centrality. Patients with inter-strain plasmid transfer may facilitate the diversification of genetic settings for $bla_{IMP-4}$, thus increasing the risk of newly successful propagators emerging[13], in turn fuelling outbreaks. Of note, we did not find clear evidence of $bla_{IMP-4}$ integron transfer events within-patients, suggesting that they may play a lesser role. Long-read technologies also allowed us to analyse putative plasmid transmission between-patients, which implicated an additional 50% of wards, 44% transmission networks and 31% patients over strain transmission alone. We used detection of $bla_{IMP-4}$ plasmids of the same MOB-typer primary/secondary clusters as a simple definition[17] but quantitative thresholds incorporating changes in plasmid backbones and large-scale recombination events across a diverse array of bacterial hosts, plasmids and resistance determinants are needed[35].

Our study had several limitations. Firstly, it was based on an isolate collection that spanned two decades with some patient data from the early part of the study being incomplete. Similarly, approaches to isolate sampling changed during that time, in particular since the inception of a statewide CPO detection programme that mandated screening in high-risk areas[36] and likely led to increased detection of CPO colonisation in the last 4 years of the study. Finally, our study focused on $bla_{IMP-4}$ and was from a single centre, which may limit the generalisability of findings to outbreaks at other centres.

In summary, we showed that $bla_{IMP-4}$ endemicity and repeated outbreaks were due to diversification of genetic settings through inter-strain $bla_{IMP-4}$ plasmid transfer and inter-plasmid $bla_{IMP-4}$ integron transfer in combination with clonal expansion that led to an evolving cascade of high-risk strain-genetic setting pairs. Our findings provide a framework for understanding endemicity of MBL-producing organisms and may have broader applicability to other CPOs. Our study highlights that stopping the spread of CPOs will require adequate surveillance to detect not only the presence of resistance determinants and their bacterial host strains but also their genetic context and plasmid-integron transmission dynamics, thus enabling early detection of novel and potentially hidden threats.

## Methods
The study was approved by the Alfred Hospital Ethics Committee (Project No: 44/20) with a waiver of consent for patient data due to its retrospective and observational nature.

### Isolate selection
We systematically reviewed an institutional collection spanning all CPO isolates from 2002 to 2020. The collection contained isolates collected as part of routine clinical care, as well as environmental screening of sinks from 2018–2020. Routine antimicrobial susceptibility testing was performed using Vitek2 (BioMérieux). We identified $bla_{IMP-4}$ carriage through polymerase chain reaction (PCR) screening.

GoTaq Flexi DNA polymerase (Promega, Wisconsin, USA) was used as per manufacturer's instructions and 10 μmol of the primers Imp4_screen_F (5'-CCAGGACACACTCCAGATAACC-3') and Imp4_screen_R (5'-CAAGAGTGATGCGTCTCCAGC-3') in 25 μL reaction volumes. PCR was performed using the following cycle conditions: 98 °C for 2 min, followed by 30 cycles of 98 °C for 30 sec, 55 °C for 30 sec, 72 °C for 30 sec. Amplicons were resolved by agarose gel electrophoresis on a 1% w/v agarose gel.

We selected 277 $bla_{IMP-4}$ isolates for whole genome sequencing (WGS) based on bacterial strain (species/MLST combination) and year of isolation. For species with <30 isolates, we sequenced all available isolates. For species with >30 isolates, we performed WGS on selected isolates based on collection date to ensure that we had sequencing data available for all study periods. We sequenced at least one isolate of all strains across all study periods with both short-read (Illumina) and long-read (Oxford Nanopore) technologies ($n = 172$). We also selected one carbapenem-susceptible *E. hormaechei* ST93 and one *E. hormaechei* ST190 isolate for short- and long-read WGS.

### Culture, DNA extraction and sequencing
All bacterial isolates were grown on cation-adjusted Mueller-Hinton II agar (Becton-Dickinson) for 16 h at 37 °C, and sub-cultured into cation-adjusted Mueller-Hinton broth (Becton-Dickinson) for a further 16 h at 37 °C. Bacterial genomic DNA was extracted from liquid culture using the GenFind V3 Reagent Kit (Beckman Coulter) as per manufacturer's instructions. Libraries for short read sequencing were prepared using the Nextera Flex DNA Library Prep Kit (Illumina), and 150 bp paired-end sequencing was performed on the NovaSeq 6000 system (Illumina). Libraries for long-read sequencing were prepared using the Ligation Sequencing Kit with Native Barcoding Expansion (Oxford Nanopore Technologies) and sequenced on the MinION instrument with an R9.4.1 flow cell (Oxford Nanopore Technologies) for 48 h. Basecalling was performed with Guppy v.4.0.14 using the 'high accuracy' basecalling model.

### De novo assembly and annotation
We constructed de novo assemblies of all isolates with only short-read data using the Shovill v1.0.4 wrapper for SPAdes, which also utilizes Trimmomatic for read trimming and Pilon for read error correction[37–40]. For long-read assembly, long reads were filtered using Filtlong v.0.2.0[41] with the following parameters: '--min_length 1000 --keep_percent 90 --target_bases 500000000'. Hybrid assemblies incorporating short- and long-read data were created using Unicycler v.0.4.08 with standard parameters[42] with Unicycler output used to assess circularisation. If $bla_{IMP-4}$ contigs were non-circularised, we re-assembled genomes using a long-read-first assembly using a bespoke pipeline (https://github.com/HughCottingham/clinopore-nf) that incorporates Flye v2.9.2 with subsequent polishing with Medaka v1.8.0, Polypolish v0.5.0 and Polca v3.4.1[43–46]. Assembly quality was checked using Quast[47] v5.2.0 and species identification was performed using GTDB-Tk[48] v1.0.2 and checked against isolate identification performed at time of isolate collection. We excluded genomes ($n = 7$) with a species mismatch, as well as genomes whose assemblies had >1000 contigs, N50 < 10,000 or assembly length >7.5 Mb.

On the remaining assemblies ($n = 270$), we annotated the genomes using Prokka v1.14.6[49]. We then performed resistance gene and plasmid replicon detection with Abricate v.1.0.0[50], using the NCBI Antibiotic Resistance and PlasmidFinder databases, respectively. We determined in silico multi-locus sequence type (ST) using 'mlst' v.2.19.0[51]. All inconclusive ST calls with 'mlst' were checked with SRST2[52] v0.2.0.

### Core genome-based phylogenetic analyses
We performed core genome-based phylogenetic analyses on key STs, defined as those with ≥5 isolates available from our

institution. This included *E. hormaechei* ST93, ST114, and ST190 and *P. aeruginosa* ST111. Due to the absence of an MLST schema for *S. marcescens* we identified all RefSeq *S. marcescens* genomes and used Assembly Dereplicator v0.1.0 (https://github.com/rrwick/Assembly-Dereplicator) with a Mash distance threshold of 0.001 to remove duplicate assemblies. We then used these assemblies, in conjunction with *S. marcescens* genomes from our institution to construct a phylogeny using Mashtree[18] v1.2.0. In brief, this tool uses non-alignment based assessment of sequence similarity through use of the min-hash algorithm, as implemented in Mash[53], to generate distance metrics between input sequences. These are then used to cluster sequences using the neighbour joining algorithm. This allowed us to identify that Alfred Hospital genomes belonged to two lineages, for which we conducted the same phylogenetic analyses as we did within STs for other species.

This consisted of identifying RefSeq genomes of the same ST and including them for context in phylogenetic analyses. We chose one completed, closed assembly from our institution for each ST to use as a reference. Mobile genetic elements were excluded from these reference assemblies using PHASTER and IslandViewer 4[54,55]. A core chromosomal SNV alignment was generated using Snippy v.4.6.0[56] and recombination was removed using Gubbins[57] v3.3. We then used this core genome alignment in IQtree v.2.0.3 to generate maximum likelihood phylogenies for each ST[58], with the best-fit model chosen using ModelFinder[59]. For each ST, median SNV distances between isolates from our institution were then calculated. Phylogenetic trees were visualized and annotated with metadata using 'ggtree'[60] with additional editing in Adobe Illustrator v2020.24.3.

### Plasmid phylogenetic analyses
Using Abricate, we identified $bla_{IMP-4}$-harbouring contigs that were putative plasmids in our hybrid assemblies. We then used the MOB-typer v1.4.9 tool to determine plasmid replicons present, as well as to assign clusters[17]. In addition, we used COPLA[61] to assign plasmid taxonomic units to key plasmid types as determined by MOB-typer. We identified possible mosaic plasmids resulting from fusion events by examining plasmid replicon content within MOB-typer cluster and identifying plasmids which had presence of additional plasmid replicons then manually inspecting the assemblies.

We then conducted analyses within key plasmid groups within our dataset, as determined by MOB-typer cluster. These included IncC, IncHI2A type 1, IncHI2A type 2, IncFIA/IncFIB/IncP, IncFIB, IncL/M, and untypeable plasmids from *Acinetobacter* spp. In order to identify SNVs in the plasmid backbone, we used Snippy v.4.6.0[56] to create a core SNV alignment by mapping short reads to a reference plasmid from our institution from each plasmid group. We then used Mashtree[18] to generate distance metrics between plasmids belonging to the same group, excluding mosaic plasmids. The R package 'ggtree' v3.0.4 was used to visualize the resulting trees[60] and to annotate with metadata. Adobe Illustrator v2020.24.3 was used to merge different parts of the figures together. We also used fastANI v1.3 to generate pairwise average nucleotide identities between plasmids belonging to the same plasmid group[62]. We used pro-gressiveMauve v2.4.0.r4736 to align all plasmids within a plasmid group and assess for structural re-arrangements[63], then visualized this in Easyfig v2.2.2[64].

### Analysis of $bla_{IMP-4}$ integron and flanking sequences
We used Flanker[19] v0.1.5 to identify and cluster flanking sequences around $bla_{IMP-4}$ from hybrid contigs. We performed clustering 5000 bp upstream and downstream of the $bla_{IMP-4}$ gene across windows in 500 bp increments. Geneious v10.2.6 (https://www.geneious.com) was used to visualize and assess for structural re-arrangements, with subsequent manual annotation in Adobe Illustrator v2020.24.3.

We also assessed for SNVs in the $bla_{IMP-4}$ integron by aligning $bla_{IMP-4}$ genetic settings from completed, circularised assemblies to a pre-viously reported $bla_{IMP-4}$ integron (GenBank accession number JX101693)[4] using MUSCLE v3.8.1551[65]. Assemblies with large scale insertions or deletions in the integron were excluded (e.g. *P. aeruginosa* ST111 genomes). We extracted SNVs from the resulting alignment using SNP-sites v2.5.1[66] and grouped plasmids according to the SNV profile.

### Short-read mapping to plasmid sequences
For genomes which only had short-read data available, we created a database of plasmids from all MOB-typer clusters (described above) and used the Nextflow implementation of the REDDog pipeline (V1.beta10.3; available at https://github.com/scwatts/reddog-nf) to map short-reads to this database. We used the following parameters: 'mapping_cover_min = 1, mapping_mapped_min = 0.5, mapping_depth_min = 10' then analysed the data. A read set was considered to have a match to a plasmid in the database if there was >90% coverage of the plasmid with <10 SNVs.

### Patient data and transmission events
Clinical data were extracted from the electronic medical record. Clinical data were missing for 7 isolates from 2009–2012. Patient movement data were available from 2013 onwards, including 127/196 (65%) patients in the study. As patients did not undergo systematic surveillance for $bla_{IMP-4}$, we considered that the patient may have been colonised in the 30 days prior to the first isolation of a $bla_{IMP-4}$-harbouring organism and identified overlaps on the same ward at the same time as potential transmission events between patients. We then applied genomic criteria to further confirm potential trans-mission events. In the first instance, patients would have to have $bla_{IMP-4}$-harbouring bacteria of the same strain for a potential trans-mission event to be considered. These events were then further classified on basis of SNV distance, with a cutoff of 10 SNVs. In the second instance, patients would have to have $bla_{IMP-4}$ in the same genetic setting (defined as the same MOB-typer primary/secondary cluster), as determined either through completed assemblies or by having a match to a reference plasmid using the short-read mapping approach described above. We then used the R package 'ggraph' v2.0.5 to visualize putative transmission networks with patients as nodes and potential transmission events as edges. Betweenness centrality was calculated using the 'betweenness' function in the 'iGraph' R package (v1.2.11)[67].

### Statistical analysis
Categorical variables were compared using χ2 or Fisher's exact tests and continuous variables were compared using Student's *t*-test or Mann–Whitney–Wilcoxon, as appropriate. Statistical analyses were performed in R (v4.1.1).

### Reporting summary
Further information on research design is available in the Nature Portfolio Reporting Summary linked to this article.

## Data availability
The Illumina/Nanopore read data generated in this study have been deposited in the NCBI Sequence Read Archive under project accession PRJNA924056. The completed genome assemblies are available in GenBank; accessions are listed in Supplementary Dataset 1. The addi-tional phylogenetic data generated in this study are provided in the Source Data file. Source data are provided with this paper.

## Code availability
The code generated during this study is available on GitHub (https://github.com/nenadmacesic/imp4_ncomms)[68].

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

## Acknowledgements
This work was supported by the National Health and Medical Research Council of Australia (Emerging Leader 1 Fellowship APP1176324 to N.M. and Practitioner Fellowship APP1117940 to A.Y.P.). The funders had no role in study design, data collection and interpretation, or the decision to submit the work for publication.

## Author contributions
N.M. and A.Y.P. conceived the study. J.A.W., A.D., D.W.S. and A.W.J.J. designed and supervised sampling and collection of bacterial isolates. L.V.B., T.H., K.P. and G.Z.B. collected the bacterial isolates, performed bacterial characterisation and conducted whole genome sequencing of isolates. N.M. collected all clinical data. N.M., J.H., B.V. and H.C. performed bioinformatics analyses. N.M., J.H. and A.Y.P. analysed all results. N.M. wrote the initial draft of the manuscript. N.M., J.H. and A.Y.P. contributed to the final version of the manuscript. All authors read and approved the manuscript.

## Competing interests
N.M. has received research support from GlaxoSmithKline, unrelated to the current study. A.Y.P. has received research funding from MSD through an investigator-initiated research project. All other authors declare no conflict of interest.
