## [Peer Review File · Nature Communications]

REVIEWER COMMENTS

Reviewer #1 (Remarks to the Author):

In this study, Macesic and collaborators investigate the dissemination of the metallo-beta-lactamase gene blaIMP-4 in a hospital over two decades. The results showed that the gene spreads in different waves. In a first wave, IncC plasmids allow the gene to disseminate and mobilize into different genetic backgrounds, including multiple different plasmids. Those plasmids allowed the further dissemination of the gene in subsequent waves. This is a really interesting and comprehensive study that captures the complexity of the epidemiology of antibiotic resistance genes in clinical settings, including the analysis of the between- and within-patient plasmid dissemination and the study of environmental reservoirs. The study is well designed and executed, the results are exciting and interpretation is clear. I have some minor comments that I enumerate below:

-My main criticism is about the figure legends. In the legends the authors discuss the results, but I miss a previous description of the panels, an explanation of what we are looking at. This affects most of the figures in the paper.

-The authors use the term “genetic settings” to refer to the different genetic backgrounds where the gene can be found (or inserted). I was wondering if the terms “genetic context” or “genetic background” would be clearer than “genetic settings” for the reader. Alternatively, the term could be defined the first time it appears in the text.

-Line 100, never on both places at the same time?

-Line 116, have the authors tested the conjugation ability of the plasmid experimentally?

-line 154-155. In the case of blaIMP-4-carrying IncHI2A plasmids, the authors were able to show that very similar plasmids without the blaIMP-4 gene were present in the hospital (lines 203-213). I was wondering if the authors have similar evidences for the IncC->IncF movement of the gene. Do the E. hormaechei ST114 strains carrying blaIMP-4 in IncC plasmids present IncF plasmids similar to those where blaIMP-4 eventually integrate?

-Line 237, “4/41 patients”, but in figure 5 I think there are 5 patients.

-lines 259-269, which two strains and plasmid types?

-line 297, maybe tone down a bit? "may also have driven" or "probably drove". I guess that, although it is quite likely, there is no clear evidence of the transfer events in the sink.

-Extended data figure 1. Could you color code by species? The genetic settings are already indicated in the panels.

Alvaro San Millan

Reviewer #2 (Remarks to the Author):

Overall a well-performed and interesting study. Lots of genomic data have been collected and analyzed, constituting an impressive amount of work.

Then, the impact of such findings still remains quite limited on the referee's opinion, considering that conclusions were rather expectable. Major mean of dissemination of a carbapenemase gene is related to clonal spread, then to plasmid exchanges and finally to gene mobilization when applicable (genes being part of mobilizable genetic structures).

Some specific remarks;

- please clarify that integrons do not move on their own, but the transposons bearing the integrons may.
- please clarify that gene cassettes such as the blaIMP-4 one might be also a source of dissemination on their own, being acquired by different class 1 integrons.
- since blaIMP-4 ended up by targeting another plasmid type, not clear this was the cassette integrating a new integron or the same integron jumping from one plasmid to another through transposition ?
- was an IncIncH2A-type plasmid found without blaIMP-4 cassette or related integron then, before being identified on this platform ?

REVIEWER 1

Comment 1:

My main criticism is about the figure legends. In the legends the authors discuss the results, but I miss a previous description of the panels, an explanation of what we are looking at. This affects most of the figures in the paper.

Response 1:

We have amended the figure legends throughout the manuscript to have more precise descriptions of figure content, including panels. We have also made the figure legends more concise, which will hopefully allow the reader to follow the figures more easily.

For example, Figure 1 legend now reads:

'Fig. 1 – Summary of bacterial host species, multi-locus sequence types and genetic settings of *bla*_{IMP-4} in sequenced isolates

Fig. 1A: Pie chart showing key bacterial host strains of *bla*_{IMP-4}. *bla*_{IMP-4} was noted in 7 bacterial genera and 68 bacterial host strains.

Fig. 1B: Genetic settings and bacterial host species of *bla*_{IMP-4} over course of study, as defined by three distinct periods.'

Please see the revised manuscript for all changes to figure legends.

Comment 2:

The authors use the term “genetic settings” to refer to the different genetic backgrounds where the gene can be found (or inserted). I was wondering if the terms “genetic context” or “genetic background” would be clearer than “genetic settings” for the reader. Alternatively, the term could be defined the first time it appears in the text.

Response 2:

We have defined ‘genetic setting’ the first time it is used in the manuscript, as suggested.

Line 52 reads:

'Previous work from our group and others have identified that the spread of MBLs, and particularly *bla*_{IMP-4}, is often driven by dissemination by transposons of a class 1 integron that has been able to insert into several genetic settings (henceforth defined as either chromosomal integration or different plasmid types carrying *bla*_{IMP-4})^{2,4,6,9,14,15}.'

Comment 3:

Line 100, never on both places at the same time?

Response 3:

This is correct, we did not detect concurrent chromosomal integration and plasmid carriage of *bla*_{IMP-4}. This detail has been added to the manuscript.

Line 105 reads:

'No genomes showed concurrent chromosomal integration and plasmid carriage of *bla*_{IMP-4}.'

Comment 4:

Line 116, have the authors tested the conjugation ability of the plasmid experimentally?

Response 4:

Unfortunately, plasmid conjugation experiments were beyond the scope of our manuscript, which focuses predominantly on the molecular epidemiology of *bla*_{IMP-4}. This is a focus of ongoing work in our group. We have removed references to plasmid promiscuousness from the manuscript.

Line 120 reads:

'The IncC plasmid was rapidly noted in diverse bacterial hosts, with expansion from *S. marcescens* into 13 other bacterial strains in 2004 alone.'

Comment 5:

Line 154-155. In the case of *bla*_{IMP-4}-carrying IncHI2A plasmids, the authors were able to show that very similar plasmids without the *bla*_{IMP-4} gene were present in the hospital (lines 203-213). I was wondering if the authors have similar evidences for the IncC->IncF movement of the gene. Do the *E. hormaechei* ST114 strains carrying *bla*_{IMP-4} in IncC plasmids present IncF plasmids similar to those where *bla*_{IMP-4} eventually integrate?

Response 5:

Thank you for the suggestion. We checked all *E. hormaechei* ST114 assemblies with MOB-typer for presence of the IncFIA/IncFIB/IncP and IncFIB plasmids. These plasmids were only present in genomes where they also carried *bla*_{IMP-4}.

Comment 6:

Line 237, "4/41 patients", but in figure 5 I think there are 5 patients.

Response 6:

This is an error from a prior draft of the manuscript with one of the patients misclassified as having persistent colonisation when they had multiple colonisation events. This has now been corrected.

Line 241 reads:

'Multiple colonisation events were noted in 5/41 patients, with *bla*_{IMP-4} being located in distinct genetic settings (i.e. differing plasmids and/or chromosomal integration).'

Line 247 reads:

'Persisting colonisation was noted in 26 patients, with the same strain and same *bla*_{IMP-4} genetic setting repeatedly isolated.'

Comment 7:

Lines 259-269, which two strains and plasmid types?

Response 7:

This patient was colonised with *E. hormaechei* ST190 with IncHI2A type 1 plasmid and *E. hormaechei* ST93 with IncHI2A type 2 plasmid. We have added this to the text.

Line 261 reads:

The patient with the highest betweenness centrality (164.0 vs mean 4.9) had a >4-month ICU admission and was implicated in 7 transmission events, placing them at the centre of the large 23-patient network spanning those two strains and plasmid types (*E. hormaechei* ST190 with IncHI2A type 1 plasmid and *E. hormaechei* ST93 with IncHI2A type 2 plasmid) (Fig. 5B).

Comment 8:

Line 297, maybe tone down a bit? "may also have driven" or "probably drove". I guess that, although it is quite likely, there is no clear evidence of the transfer events in the sink.

Response 8:

We have made the changes as requested.

Line 301 reads:

'This environmental colonisation probably enabled clonal spread of *bla*_{IMP-4} harbouring *S. marcescens* and may have driven diversification through inter-strain transfer of *bla*_{IMP-4} IncC plasmids and inter-plasmid transfer of the *bla*_{IMP-4} class I integron.'

Comment 9:

Extended data figure 1. Could you color code by species? The genetic settings are already indicated in the panels.

Response 9:

This has been done as requested. Please see Extended Data Fig. 1 in the manuscript.

REVIEWER 2**Comment 1:**

The impact of such findings still remains quite limited on the referee's opinion, considering that conclusions were rather expectable. Major mean of dissemination of a carbapenemase gene is related to clonal spread, then to plasmid exchanges and finally to gene mobilization when applicable (genes being part of mobilizable genetic structures).

Response 1:

We had a unique dataset of two decades of *bla*_{IMP-4} carbapenemases (including both clinical and environmental isolates) and used multiple sequencing technologies to characterise the genetic settings of these carbapenemase genes in a systematic manner, while tightly linking this to clinical epidemiological data. The findings of multiple mechanisms of spread being present including strain transmission, plasmid transmission and transfer of the *bla*_{IMP-4} class 1 integron are novel. Prior studies have mostly captured datasets where a single such mechanism acts at any given time (eg. León-Sampedro, R. *et al. Nat Microbiol* (2021) doi:10.1038/s41564-021-00879-y; David, S. *et al. Proc National Acad Sci* (2020) doi:10.1073/pnas.2003407117). We feel that these findings are impactful and carry important implications for future infection prevention and surveillance efforts and provide a model for how carbapenemase endemicity may be established.

Comment 2:

Please clarify that integrons do not move on their own, but the transposons bearing the integrons may.

Response 2:

We have tried to address this at important junctures in the manuscript.

Line 52 reads:

'Previous work from our group and others have identified that the spread of MBLs, and particularly *bla*_{IMP-4}, is often driven by dissemination by transposons of a class 1 integron that has been able to insert into several genetic settings (henceforth defined as either chromosomal integration or different plasmid types carrying *bla*_{IMP-4})^{2,4,6,9,14,15}.'

Line 281 reads:

'Firstly, there is a need for diversification of genetic settings for the resistance determinant, which in our study occurred through extensive inter-strain transmission of key *bla*_{IMP-4} plasmids (IncC, IncHIA2 type 1 and type 2, IncL/M), as well as mobilisation by transposons of the class I integron and entry into new plasmids (IncFIB, IncFIA/IncFIB/IncP).'

Comment 3:

Please clarify that gene cassettes such as the *bla*_{IMP-4} one might be also a source of dissemination on their own, being acquired by different class 1 integrons.

Response 3:

We have added this to the manuscript and have provided an appropriate reference.

Line 55 reads:

'Furthermore, the gene cassettes (such as that carrying *bla*_{IMP-4}) may also be a source of dissemination by being able to enter different class 1 integrons¹⁶.'

Comment 4:

Since *bla*_{IMP-4} ended up by targeting another plasmid type, not clear this was the cassette integrating a new integron or the same integron jumping from one plasmid to another through transposition?

Response 4:

It was the same class I integron moving from one plasmid to another through transposition. We refer the reviewer to Figure 3B, where we performed a clustering analysis of flanking regions of *bla*_{IMP-4}. The same *IntI* and cassette array containing *bla*_{IMP-4} was present in all plasmids shown in this figure but the flanking regions (including the transposons) differed between them. IncC, IncFIA/FIB/IncP and IncFIB plasmids shared the same Tn3 transposons and DNA recombinases (*hin*) located immediately upstream (as stated on line 166), which were different to the regions upstream of the class I integron in IncHI2A plasmids and IncL/M plasmids.

Comment 5:

Was an IncH2A-type plasmid found without *bla*_{IMP-4} cassette or related integron then, before being identified on this platform?

Response 5:

This is correct – we identified contemporary *E. hormaechei* ST114, ST190 and ST93 isolates from our institution, which we then sequenced (see line 208). These isolates had the same IncHI2A plasmids (IncHI2A type 1 and IncHI2A type 2, respectively) but did not have the *bla*_{IMP-4} cassette (see Figures 4B and 4C).